# C5, A Cassaine Diterpenoid Amine, Induces Apoptosis via the Extrinsic Pathways in Human Lung Cancer Cells and Human Lymphoma Cells

**DOI:** 10.3390/ijms21041298

**Published:** 2020-02-14

**Authors:** Hyo-Jin Kim, Bo-Gyeong Seo, Kwang Dong Kim, Jiyun Yoo, Joon-Hee Lee, Byung-Sun Min, Jeong-Hyung Lee, Cheol Hwangbo

**Affiliations:** 1Division of Life Science, College of Natural Sciences, Gyeongsang National University, Jinju 52828, Korea; jin4477@hanmail.net (H.-J.K.); sbk6427@naver.com (B.-G.S.); kdkim88@gnu.ac.kr (K.D.K.); yooj@gnu.ac.kr (J.Y.); 2Division of Applied Life Science (BK21 Plus), PMBBRC and Research institute of Life Sciences, Geongsang National University, Jinju 52828, Korea; 3Department of Animal Bioscience, College of Agriculture and Life Sciences, Gyeongsang National University, Jinju 52828, Korea; sbxjhl@gnu.ac.kr; 4College of Pharmacy, Catholic University of Daegu, Daegu 38430, Korea; bsmin@cu.ac.kr; 5Department of Biochemistry, College of Natural Sciences, Kangwon National University, Chuncheon 24341, Korea

**Keywords:** C5 (3β-Acetyl-nor-erythrophlamide), apoptosis, extrinsic pathways, Bcl-2, caspase-8

## Abstract

Apoptosis pathways in cells are classified into two pathways: the extrinsic pathway, mediated by binding of the ligand to a death receptor and the intrinsic pathway, mediated by mitochondria. Apoptosis is regulated by various proteins such as Bcl-2 (B-cell lymphoma 2) family and cellular FLICE (Fas-associated Death Domain Protein Interleukin-1β-converting enzyme)-inhibitory protein (c-FLIP), which have been reported to inhibit caspase-8 activity. In this study, it was found that C5 (3β-Acetyl-nor-erythrophlamide), a compound of cassaine diterpene amine from *Erythrophleum fordii*, induced cell apoptosis in a variety of types of cancer cells. Induction of apoptosis in cancer cells by C5 was inversely related to the level of Bcl-2 expression. Overexpression of Bcl-2 into cancer cells significantly decreased C5-induced apoptosis. It was also found that treatment of cancer cells with a caspase-8 inhibitor significantly suppressed C5-induced apoptosis; however, treatment with caspase-9 inhibitors did not affect C5-induced apoptosis, suggesting that C5 may induce apoptosis via the extrinsic pathway by activating caspase-8. It was confirmed that treatment with C5 alone induced an association of FADD with procaspase-8; however, overexpression of c-FLIP decreased C5-induced caspase-8 activation. In conclusion, C5 could be utilized as a new useful lead compound for the development of an anti-cancer agent that has the goal of apoptosis.

## 1. Introduction

Apoptosis is the process of programmed cell death, generally known as morphological and biological modifications, including cell shrinkage, nuclear fragmentation, and membrane blebbing [1,2]. There are two distinct mechanisms of apoptosis: intrinsic and extrinsic pathways [3]. Stress such as radiation, hypoxia, viral infection, and increased concentration of intracellular calcium activate the intrinsic pathway [4,5]. This pathway is mediated by Bcl-2 (B-cell lymphoma 2) family proteins, including pro-apoptotic proteins (such as Bax, Bak, Bad, and Bid) and anti-apoptotic proteins (such as Bcl-2, Bcl-XL, and Mcl-1) [6]. Increased expression of Bax/Bak proteins is caused by cytochrome c secretion and released cytochrome c interacts with apaf-1, forming a protein complex known as apoptosome via pro-caspase-9 binds, then activating pro-caspase-9 by apoptosome as caspase-9, followed by caspase-3 activation causing apoptosis [7,8,9]. The extrinsic pathway is activated by binding of death receptor and death ligand as TNFR (tumor necrosis factor receptor) and Fas (CD95) and their TNF and FasL (Fas ligand) ligands, respectively [10,11]. The receptor and ligand binding activate caspase proteins through intracellular domains, such as the TNF receptor-associated death domain (TRADD) and Fas-associated death domain (FADD). In the Fas pathway, the Fas receptor and ligand binding cause a formation of death-inducing signaling complex (DISC), including FADD, caspase-8, and caspase-10 [12]. The FADD-like interleukin-1β–converting enzyme (FLICE)-inhibitory protein (c-FLIP) is an anti-apoptotic regulator, it inhibits binding between caspase-8 and FADD by interaction with FADD [13]. Therefore, DISC activates caspase-8, which is the apoptosis initiator protein, through pro-caspase-8 cleavage and promotes apoptosis execution [14].

C5 (3β-Acetyl-nor-erythrophlamide) is one of the mono cassaine diterpenoid amides derived from *Erythrophleum fordii*, which has numerous pharmacological effects including angiogenesis on human umbilical vascular endothelial cells (HUVECs) and apoptosis-inducing activity against certain cancer cell lines [15,16,17]. However, the biological functions of their underlying molecular mechanisms of the anti-tumor activity remain unclear. Therefore, the objective of the present study aimed to determine the effect of C5 on apoptosis in a variety of types of cancer cells.

## 2. Results

### 2.1. C5 inhibits Cell Proliferation in Human Cancer Cell Lines

In order to evaluate the anti-cancer effect of C5 in a variety of types of cancer cells, including hematological and solid human cancer cell lines, an indicated concentration of C5 was administered to the human lung cancer cell lines (A549 and NCI-H1299), lymphoma cell lines (Ramos, U937 and Daudi), liver cancer cell line (Huh7), and prostate cancer cell (PC3) for 48 h, followed by the MTT assay. While in most lymphoma cell lines C5 had little effect on survival, it significantly reduced the viability of Ramos cells by about 74.4% at 0.3 uM (Figure 1A). Furthermore, C5 also reduced the viability of A549 cells by about 77.8% at 0.3 uM higher than other cancer cell lines (Figure 1B), indicating that C5 could inhibit cell proliferation in some human cancer cells.

### 2.2. C5 Induces Apoptosis in the Ramos and A549 Cells

As a result of the effects described, C5 significantly reduced cell viability of Ramos cells in a concentration-dependent manner compared to other lymphoma cell lines (Figure 1A). To further examine whether C5 exert anti-cancer activity by inducing apoptosis in human lymphoma cell lines and lung cell lines, evaluating C5-inducing apoptosis activity, the cancer cells were treated with C5 in a concentration- or time-dependent manner and apoptosis was analyzed using Annexin-V/PI double staining (Figure 2). Treatment of Ramos cells with C5 has increased the population of Annexin-V^+^/PI^+^ (late apoptosis) and Annexin-V^+^/PI^−^ (early apoptosis) cells in a concentration-dependent manner, whereas the population of Annexin-V^−^/PI^+^ (necrosis) has not been shown (Figure 2A). However, Daudi cell treatment with C5 did not significantly increase the population of Annexin-V^+^/PI^+^ (late apoptosis) and Annexin-V^+^/PI^−^ (early apoptosis) cells, indicating that C5 more effectively induces apoptosis in Ramos cells than Daudi cells (Figure 2A). In addition, a time-dependent treatment of Ramos cells with C5 increased the population of Annexin-V^+^/PI^+^ (late apoptosis) and Annexin-V^+^/PI^−^ (early apoptosis) cells as a concentration-dependent manner but not in Daudi cells (Figure 2B). Similar to the effect in Ramos cells, treatment of A549 cells with C5 significantly increased the population of Annexin-V^+^/PI^+^ (late apoptosis) and Annexin-V^+^/PI^−^ (early apoptosis) cells, but less so in NCI-H1299 cells (Figure 2C). As apoptotic marker proteins, the cleavage of PARP and capase-3 were measured by western blot analysis in Ramos cells and A549 cells in a C5 concentration-dependent manner. Treatment of Ramos cells and A549 cells with C5 increased the levels of cleaved PARP and caspase-3 in a concentration-dependent manner (Figure 2D, E), indicating that it could induce apoptosis in Ramos and A549 cells via caspase pathways.

### 2.3. Bcl-2 Expression Level Affects C5-Induced Apoptosis Sensitivity

Western blot analysis revealed that the cells resistant against C5-induced apoptosis, such as U937, Daudi, and NCI-H1299 cells, have high levels of Bcl-2 expression in contrast with Ramos cells and A549 cells, which well induced apoptosis by C5 (Figure 3A). In Figure 2A–C, the Daudi and NCI-H1299 cells were less affected by C5 than the Ramos and A549 cells. Bcl-2 protein expression is therefore expected to be a critical molecule for apoptosis induced by C5. We have identified whether C5-induced cell apoptosis in A549 was repressed by Bcl-2 expression with a flow cytometry analysis (Figure 3B). C5-inducing apoptosis was inhibited by Flag-Bcl-2 overexpression in A549 cells, and such findings indicate that Bcl-2 is likely to be susceptible to C5-induced apoptosis in cancer cells.

### 2.4. Caspase-8 Activity Essential for C5-Induced Apoptosis

To further determine whether C5 induces an intrinsic or extrinsic pathway in inducing apoptosis, we measured caspase-8 cleavage, Bid, caspase-3 cleavage, and PARP using western blot analysis. Once C5 was treated for 12 h, caspase-8 cleavage, a key element of extrinsic pathway apoptosis, was increased, followed by caspase-3 cleavage (Figure 4A). Next, we performed flow cytometry analysis with caspase-8 inhibitor or caspase-9 inhibitor to confirm whether C5-induced apoptosis is via the extrinsic pathway (Figure 4B). In Ramos cells, Z-IETD-FMK (caspase-8 inhibitor) significantly inhibited C5-induced apoptosis but not Z-LEHD-FMK (caspase-9 inhibitor). In addition, C5 treatment in A549 cells increased caspase-8 cleavage, and Z-IETD-FMK effectively suppressed C5-induced apoptosis (Figure 4C). These data indicate that apoptosis induced by C5 is via the extrinsic pathway.

### 2.5. C5 Induces the Formation of DISC without Affecting the Expression of Death Receptor and Death Ligand in Extrinsic Apoptosis Pathway

To evaluate the molecular mechanism of the extrinsic pathway regulated by C5, we investigated the expression of the death receptor and death ligand that activate caspase-8. Fas receptor expression was not affected by C5 in response to a time-dependent manner in Ramos cells (Appendix A). Also, cell surface expression levels of the Fas receptor were measured using flow cytometry analysis, but cell surface Fas receptor expression was not significantly altered by C5 (Appendix A). The mRNA expression of various death ligands, such as death receptor 4 (DR4), TNF-related apoptosis-inducing ligand (TRAIL), and FasL, were measured in response to C5 (Appendix A). However, the mRNA levels of DR4, TRAIL, and FasL were not altered by C5 at any indicated time, indicating that C5 had no effect on the expression of death receptors and death ligands; therefore, we investigated whether C5 affected the formation of DISC. The interaction between the death receptor and the death ligand enhances DISC formation, which causes the activation of caspase-8 [14]. To determine whether C5 increases DISC formation, we conducted an immunoprecipitation assay and found that treatment of C5 in Ramos cells increases the interactions between caspase-8 and FADD (Figure 5A). Next, we examined whether the overexpression of c-FLIP could reduce the cleavage of caspase-8 caused by C5 (Figure 5B). The flag-c-FLIP overexpression in Ramos cells decreased C5-induced caspase-8 cleavage. Taken together, C5 led to the formation of DISC to activate caspase-8 which induces the process of the extrinsic apoptosis pathway. 

## 3. Discussion

Natural products are less harmful to the human body than synthetic drugs [18,19], and because of these properties, they are used to treat various diseases, including cancer [20,21]. *Erythrophleum fordii* belongs to the subfamily of Caesalpinioideae and the Leguminosae family [22]. *Erythrophleum fordii* Oliv is widely distributed in China, Taiwan, and Vietnam [23]. This plant is used for the improvement of blood circulation in Chinese oriental medicine [24]. Various chemicals are derived from *Erythrophleum fordii*, such as cassaine ditetpenoids, diterpenes, triterpenoids, and flavanonol [25,26,27,28]. C5 is one of the natural extracts of *Erythrophleum fordii* and one form of cassaine diterpene alkaloid, also known as 3β-acetyl-nor-erythrophlamide [29]. Pharmacologically, the Cassaine diterpene alkaloid influences the function of cytotoxicity, inflammation, and angiogenesis activity [17,18,19,29,30]. The 3β-acetyl-nor-erythrophlamide induces apoptotic activity in human prostate cancer cells and has an anti-angiogenesis effect in human umbilical vein endothelial cells (HUVECs) [17,31]. However, a clear effect mechanism of C5 in human cancer cells was not identified.

Apoptosis is the physiological process of cell suicide, and these cells undergo morphological and biochemical changes [1]. Apoptotic cells undergo biochemical modifications, such as DNA degradation, cell membrane alterations, and activation of caspase proteins [32]. Apoptosis is accelerated via an intrinsic or extrinsic pathway [1]. The intrinsic pathway is commonly referred to as a mitochondrial pathway because they release cytochrome c into cytoplasm from mitochondria and induce mitochondrial permeability [33]. This pathway is stimulated by a number of stresses, such as DNA damage, hypoxia, viral infection, and concentrations of cytosolic Ca^2+^ [4,34]. The intrinsic pathway is controlled by Bcl-2 family proteins, including anti-apoptotic ones that inhibit apoptosis and pro-apoptotic ones that facilitate apoptosis [6,35]. These proteins regulate the release of cytochrome c, so the balance of anti- and pro-apoptotic proteins is essential for an intact apoptosis process [6,35]. Cytochrome c secretion leads to the formation of a complex called apoptosome between cytochrome c, Apaf-1, and caspase-9, and induces the activation of caspase-3, which is the initiator of apoptosis [8]. The extrinsic pathway is activated by the binding of death receptors, such as TNFR and Fas, and death ligands, such as TNF and FasL [10,11]. The interaction of death receptors and ligands recruit downstream molecules through TRADD and FADD and subsequently induce the formation of DISC [12,36]. DISC increases the activation of pro-caspase-8 by cleavage, then activates other caspase proteins and initiates apoptosis [37,38].

Malignant cancer is caused by multiple cell variations, one of which is the avoidance of apoptosis [39]. Disintegration of apoptosis is caused by an imbalance of pro-apoptotic and anti-apoptotic proteins, reduced expression of caspase, and a loss of death receptor signaling [40]. For pro-apoptotic protein, Bax induces apoptosis and decreases drug resistance in hepatocellular cancer cells [41]. Bax deficiency is associated with anticancer drug resistance in colorectal cancer cells [42]. On the other hand, overexpression of Bcl-2 protein, which is anti-apoptotic protein, contributes to the evasion of apoptosis in prostate cancer cells, glioblastoma, and breast carcinoma cells [43,44]. Bcl-xL overexpression also promotes cancer cell survival and reduces various drug resistance [45,46].

In this study, we found that C5 promotes cell apoptosis and suppresses growth of cancer cells. However, these results for the C5 effect were inconsistent in a variety of types of cancer cell lines, and then the expression of Bcl-2 family protein was investigated to determine the cause. The cancer cell lines that were weakly affected by C5 had high Bcl-2 expression levels and those that were strongly affected by C5 had low Bcl-2 expression levels. In order to investigate which pathways are relevant for C5-induced apoptosis, we measured the caspase protein activity by western blotting and found that caspase-8 cleavage is increased, which is crucial for extrinsic apoptosis pathways. Furthermore, Z-IETD-FMK, a caspase-8 inhibitor, is significantly suppressed by C5-induced apoptosis, but not by C-LEHD-FMK, a caspase-9 inhibitor in Ramos cells and NCI-H1299. Then, we checked the death receptor and ligand expressions, but no changes in expression were detected, so we assumed that C5-induced apoptosis was mediated by an extrinsic pathway without regulating death receptor and ligand expressions. Finally, we found that C5-treated cells increased the binding of FADD and caspase-8 that are the DISC components, and c-FLIP, the DISC compound inhibitor protein, reduced apoptosis in C5-induced apoptosis.

A lot of research has suggested that the natural products have the possibility to act as anti-cancer drugs. Substances are commonly known to affect cancer cells by controlling apoptosis-related proteins, such as NK-κB and p53, rather than directly controlling the cell apoptosis pathway [47]. Knowing the action mechanism of certain substances within cells is very important in developing new drugs. We have revealed that C5-induced apoptosis of cancer cells occurs via extrinsic pathways and regulates the formation of the DISC complex. These findings provide a detailed mechanism for C5-induced apoptosis and strong evidence that it might be a potential candidate for the treatment of cancer drug therapy.

## 4. Material and Method

### 4.1. Cell Culture and Reagents

Human lung cancer cell lines, A549 and NCI-H1299, human lymphoma cell lines, Ramos, U937, and Daudi, human liver cancer cell line, huh7, and human prostate cancer cell line, PC3, were obtained from the American Type Culture Collection (Manassas, VA, USA) and cultured in RPMI-1640 medium (Hyclone, Logan, UT, USA) supplemented with 10% heat-inactivated fetal bovine serum (FBS, Hyclone, Logan, UT, USA), 100 U/mL penicillin, and 100 μg/mL streptomycin (Invitrogen, Waltham, Massachusetts, USA) in a humidified 37 °C incubator with 5% CO_2_. Caspase-8 inhibitor, Z-IETD-FMK, and caspase-9 inhibitor, Z-LEHD-FMK, were purchased from BD Bioscience (San Jose, USA).

### 4.2. Cell Cytotoxicity Assay

Cell proliferation was determined by direct cell counting with a hemocytometer and the MTT assay was done. In brief, cancer cells were suspended with trypsin/EDTA (Invitrogen, Waltham, Massachusetts, USA) and seeded into a 96-well plate (4 × 10^3^ cells/well), and then incubated with the indicated concentrations of C5 for 48 h. For the MTT assay, 5 mg/mL MTT was added to each well for 2 h. The insoluble formazan products were then dissolved in DMSO and absorbance was determined at 570 nm using an ELISA reader (BioTek, Power Wave XS2, Winooski, Vermont, USA).

### 4.3. Western Blotting and Antibodies

Cells were harvested and lysed in cell lysis buffer (50 mM Tris-HCl (pH 7.5), 150 mM NaCl, 1% Nonidet P-40, 1 mM EDTA, 5 mM sodium orthovanadate, 1X protease inhibitor cocktail (BD Bioscience, San Jose, USA)). Equal amounts of proteins were separated by SDS-polyacrylamide gel electrophoresis (SDS-PAGE) and then transferred onto a PVDF membrane (Amersham biosciences, Waltham, Massachusetts, USA). The membrane was blocked with 5% skim milk (BioShop, Burlington, Canada) for 1 h at room temperature (RT), then probed with the appropriate primary antibodies. Antibodies against BID, BAX, Bcl-2, Bcl-XL, and FADD were purchased from Santa Cruz Biotechnology (Santa Cruz, CA, USA). Antibodies against Caspase-3, Caspase-8, PARP, horseradish peroxidase (HRP) conjugated anti-rabbit, anti-mouse, and anti-rat antibodies were purchased from Cell Signaling Technology (Beverly, USA). Antibody against α-tubulin was purchased from Sigma-Aldrich, and Fas/CD95 was purchased from Proteintech, and Flag was purchased from Stratagene. The secondary antibodies were diluted at a 1:2000 ratio in 5% skim milk and incubated for 3 h at RT. Chemiluminescence signals were detected by a chemiluminescent system (Intron).

### 4.4. Vector and DNA Transfection

The expression vectors for FLIP and Bcl-2 were amplified by polymerase chain reaction, digested with appropriate restriction enzymes, and cloned in-frame into a pCMV-Tag 2B vector (Agilent Technology). Then, the constructs were confirmed by DNA-sequencing. Transfections were performed using Lipofectamine 3000 reagent according to the manufacturer’s instructions (Invitrogen, Waltham, Massachusetts, USA). 

### 4.5. Immunoprecipitation Assay

Immunoprecipitation and Western blotting were described previously [16]. Briefly, cells were lysed in lysis buffer (50 mM Tris-HCl (pH 7.5), 150 mM NaCl, 1% Nonidet P-40, 1 mM EDTA, 5 mM sodium ortho vanadate, 1× protease inhibitor cocktail (BD)), and centrifuged at 15,000 rpm for 30 min. For immunoprecipitation, equivalent amounts of cell lysates were incubated with the appropriate antibodies, followed by incubation protein A/G agarose bead. Immunoprecipitates were extensively washed, resolved by SDS-PAGE, transferred to PVDF membrane, and probed with appropriate antibodies. The signals were detected using an enhanced chemiluminescent system (Intron, Seongnam-si, Republic of Korea).

### 4.6. Apoptosis Analysis

Apoptosis analysis was performed using an Annexin-V/PI apoptosis detection kit according to the manufacturer’s instructions (BD Bioscience, San Jose, USA). Briefly, cells were washed 2 times with PBS and resuspended in annexin V binding buffer. Annexin V and propidium iodide (PI) reagent were added in the cells and incubated for 15 min in blocked light at RT. The cells were analyzed by flow cytometer (BD Bioscience, San Jose, USA).

### 4.7. Analysis of the Cell Surface Expression of Death Receptors

Cells were washed twice with PBS and centrifuged. Fas antibody was added in cells and incubated 30 min at RT. After washing 3 times with FACS buffer (1% BSA and 0.1% sodium azide in PBS), Alexa flour 488 second antibody was added and incubated for 30 min at RT in blocked light. After washing 3 times with FACS buffer, fluorescent was analyzed with a flow cytometer (BD Bioscience, San Jose, USA) to confirm receptor expression on the cell surface.

### 4.8. Reverse Transcription-PCR Analysis 

Total RNA was isolated using a RNeasy mini kit (Qiagen, Valencia, CA, USA) according to the manufacturer’s instructions. Briefly, cells were lysed with the buffer RLT and 1 volume of 70% ethanol was added to the lysate by mixed pipetting. All lysate was transferred to the RNeasy spin column, centrifuged, and the flow-through was discarded. RW1 buffer was added to the spin column, centrifuged, and the flow-through was discarded. RPE buffer was added, and the previous step was repeated. The buffer was transferred to a new collection tube and the RNA was eluted using RNase free water. cDNA synthesis was performed using Maxime RT PreMix (Intron, Seongnam-si, Republic of Korea) according to the manufacturer’s protocol. Appropriate amounts of cDNA, primer, and TOPreal™ qPCR 2× PreMIX (Enzynomics, Daejeon, Republic of Korea) were mixed and measured using a real-time PCR cycler (Qiagen, Valencia, CA, USA)) instrument.

The primers used can be described as follows: DR4: 5′- ACC TTC AAG TTT GTC GTC GTC-3′ (forward), 5′- CCA AAG GGC TAT GTT CCC ATT-3′ (reverse), FasL: 5′- ATT TAA CAG GCA AGT CCA ACT CA-3′ (forward), 5′- GGC CAC CCT TCT TAT ACT TCA CT-3′ (reverse), TRAIL: 5′- TGC GTG CTG ATC GTG ATC TTC-3′ (forward), 5′- GCT CGT TGG TAA AGT ACA CGT A-3′ (reverse).

### 4.9. Statistical Analysis

GraphPad prism version 7.01 (GraphPad software) was used for data analysis. All experiments were performed in triplicate, and all data are shown means ± SEM. Statistical significance was assessed by one-way analysis of variance (ANOVA) and Tukey’s test in order to examine differences between the two groups using SPSS version 14.0 (SPNN Inc., Chicago, IL, USA). *p*-values of less than 0.05 were considered statistically significant.

## 5. Conclusions

In this study, we identified the anticancer mechanism of C5, a natural substance derived from *Erythrophleum fordii*. We have confirmed that the effect of C5 is better in cancer cell lines where the high expression of Bcl-2 than low expression cell lines. Through caspase-8 and -9 inhibitor, it was confirmed that C5 induced apoptosis through extrinsic pathway and induced the combination of FADD and caspase-8 during the process to facilitate the formation of the DISC complex. We suggest that the new acting mechanism of C5 at caccer cells.

## Figures and Tables

**Figure 1 ijms-21-01298-f001:**
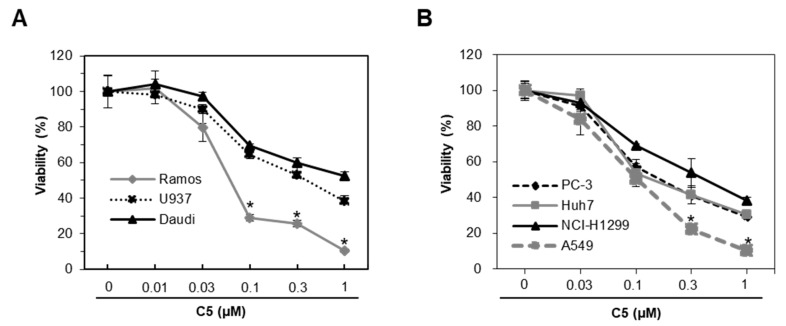
Effect of C5 in viability of cancer cell lines. (**A**) Ramos, U937, and Daudi cell lines were plated in 96-well plates and incubated in the presence of indicated concentrations of C5 for 48 h. Cell viability was measured by MTT assay. Data are presented as the mean ± SEM (* *p* < 0.01, versus Daudi; *n* = 3). (**B**) PC-3, Huh7, NCI-H1299, and A549 cell lines were plated in 96-well plates and incubated in the presence of indicated concentrations of C5 for 48 h. Cell viability was measured by MTT assay. Data are presented as the mean ± SEM (* *p* < 0.01, versus NCI-H1299; *n* = 3).

**Figure 2 ijms-21-01298-f002:**
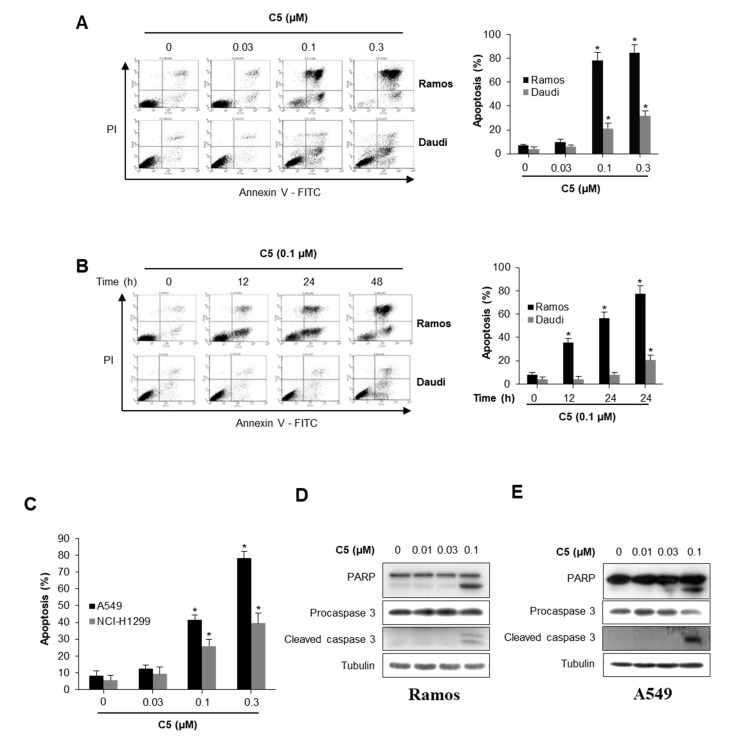
C5 induces apoptosis in cancer cells. (**A**) Ramos and Daudi cell lines treated at the indicated concentration of C5 for 48 h and subsequently stained with Annexin V and PI, followed by analysis using flow cytometry. Representative dot plots and graph are shown. Data are presented as the mean ± SEM (* *p* < 0.01, versus vehicle-treated control; *n* = 3). (**B**) Ramos and Daudi cell lines treated at the indicated time, with 0.1 μM of C5 and subsequently stained with Annexin V and PI, followed by analysis using flow cytometry. Representative dot plots and graph are shown. Data are presented as the mean ± SEM (* *p* < 0.01, versus vehicle-treated control; *n* = 3). (**C**) A549 and H1299 cell lines treated at the indicated concentration of C5 for 48 h and subsequently stained with Annexin V and PI, followed by analysis using flow cytometry. The percentage of apoptotic cells is shown in the bar graph. Data are presented as the mean ± SEM (* *p* < 0.01, versus vehicle-treated control; *n* = 3). (**D**) PARP and caspase-3 protein level in Ramos cells in response to the indicated C5 concentration for 12 h. Whole cell lysates were blotted with the indicated antibodies. (**E**) PARP and caspase-3 protein level in A549 cells in response to the indicated C5 concentration for 24 h. Whole cell lysates were blotted with the indicated antibodies.

**Figure 3 ijms-21-01298-f003:**
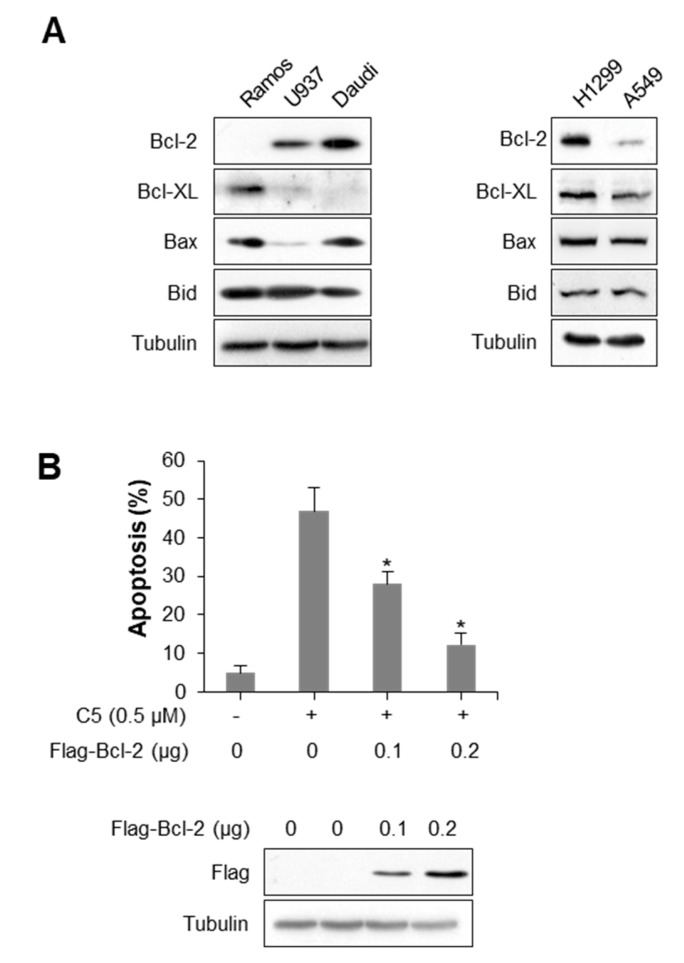
Bcl-2 expression is essential for C5-induced apoptosis. (**A**) Bcl-2, Bcl-xL, and Bax protein expression levels in lymphoma and lung cancer cell lines. Whole cell lysates were blotted with the indicated antibodies. (**B**) Flow cytometry assay for apoptosis in response to C5 treatment in flag-Bcl-2-transfected A549 cells. A549 cells were transfected with the indicated amount of flag-Bcl-2 and treated with 0.5 μM of C5 for 48 h and subsequently stained with Annexin V and PI, followed by analysis using flow cytometry. The percentage of apoptotic cells is shown in the bar graph. Data are presented as the mean ± SEM (* *p* < 0.01, versus vehicle-treated control; *n* = 3). Whole cell lysates were blotted with the indicated antibodies.

**Figure 4 ijms-21-01298-f004:**
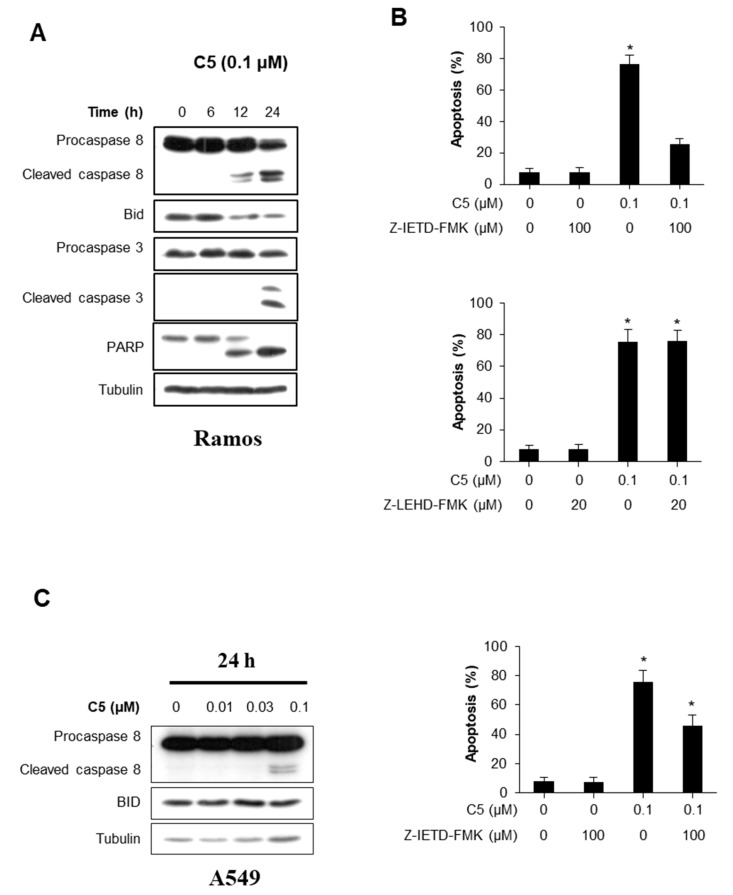
C5 induced apoptosis through caspase-8 activation. (**A**) Ramos cells were treated with 0.1 uM of C5 for the indicated time. Whole cell lysates were blotted with the indicated antibodies. (**B**) Ramos cells were treated with 0.1 uM of C5 with or without caspase-8 inhibitor (Z-IETD-FMK) or caspase-9 inhibitor (Z-LEHD-FMK), and subsequently stained with Annexin V and PI, followed by analysis using flow cytometry. Data are presented as the mean ± SEM (* *p* < 0.01, versus vehicle-treated control; *n* = 3). (**C**) Western blotting for caspase-8 cleavage in A549 cells in response to C5 does-dependent manners. Data are presented as the mean ± SEM (* *p* < 0.01, versus vehicle-treated control; *n* = 3).

**Figure 5 ijms-21-01298-f005:**
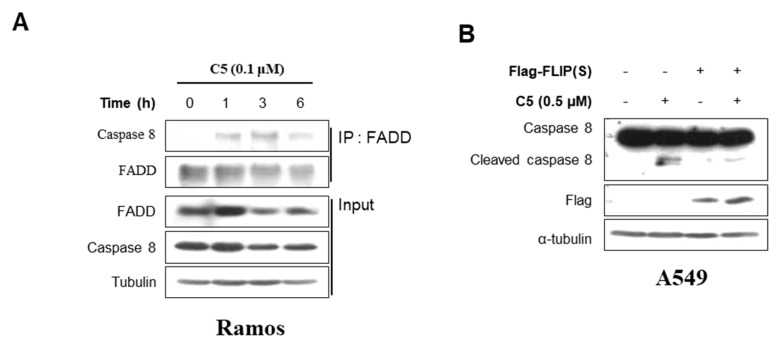
C5 affects the formation of DISC. (**A**) Co-immunoprecipitation assay between FADD and caspase-8 in Ramos cells treated with 0.1 μM C5 at the indicated time. (**B**) Western blot for caspase-8 when 0.5 μM C5 is treated in A549 cells and flag-FLIP overexpressing A549 cells. Whole cell lysates were blotted with the indicated antibodies.

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
