# Peer review of "C5, A Cassaine Diterpenoid Amine, Induces Apoptosis via the Extrinsic Pathways in Human Lung Cancer Cells and Human Lymphoma Cells"

_ijms, 2020, doi:10.3390/ijms21041298_

Round 1

Reviewer 1 Report

This is an overall well written manuscript with shows good results. It is interesting for a wider audience. The concept is good and the paper is well organized, the results are depicted in an adequate way.

Overall, I have no major concern, but here are a number of suggestions to improve the manuscript:

There are a number of typos (e.g. commas between the numbers of the literature, see line 35, 44 or 262), the statement that natural products are less harmful (line 252) needs clarification.

The discussion misses to discuss whether there are any other (natural) substances which interfere with the same pathway.

The discussion would be strongly improved by a summarizing figure.

I suggest to move the first chapter of the discussion to the introduction. 

Author Response

We thank you for allowing us to submit to International Journal of Molecular Sciences a revised draft of our manuscript entitled, “C5, a cassaine diterpenoid amine, induces apoptosis via the extrinsic pathways in human lung cancer cells and human lymphoma cells.”. We also appreciate you and each reviewer for the time and effort devoted to providing valuable suggestions on how to improve our manuscript. We have reflected the detailed proposals you have generously provided. We also hope that the following edits and answers we provide will be satisfactorily addressed to all the issues and concerns you and the reviewer have pointed out.

To facilitate the review of our revisions, the following is a point-by-point response to the questions and comments.

This is an overall well written manuscript with shows good results. It is interesting for a wider audience. The concept is good and the paper is well organized, the results are depicted in an adequate way.

We are very grateful for your time and effort to evaluate our manuscript. We tried to ascertain the exact mechanism of C5 that is expected to work in curbing cancer. We have identified that C5-induced apoptosis of cancer cells occurs via extrinsic pathways. We appreciate your assessment that our results are meaningful.

Overall, I have no major concern, but here are a number of suggestions to improve the manuscript:

We thank you for your valuable reviews, and tried to respond to your comments and suggestions. We hope the following answers are sufficient for you.

There are a number of typos (e.g. commas between the numbers of the literature, see line 35, 44 or 262),

We thank you for pointing out a number of typos we didn't recognize. We revised the manuscript reflecting your comments.

the statement that natural products are less harmful (line 252) needs clarification.

We agree with you and have clarified that natural products are less harmful as adding references by line 262. These research show that natural substances are relatively less toxic to normal cells than artificial chemicals.

The discussion misses to discuss whether there are any other (natural) substances which interfere with the same pathway.

We appreciate your opinion for our discussion. We agree with you and have incorporated this suggestion in manuscript by line 304. We tried to reflect your proposal and added a reference on the mechanism in which natural substances act as an anti-cancer effect in cancer cells. As your suggested, the difference between these references and our manuscript will make our manuscript more meaningful.

The discussion would be strongly improved by a summarizing figure.

We appreciate your opinion for our discussion. You have raised an important point, based on the proposal of a new drug candidate for anticancer, and discovered the mechanism of action of the new substance. As you suggest that the discussion has been modified to explain the C5 mechanism strongly. We believe that our discussion fully explains the mechanism of C5, and its potential as an anti-cancer drug.

I suggest to move the first chapter of the discussion to the introduction.

We appreciate your suggestion of improvement about our manuscript. However, we focused on the working mechanism of C5 in cancer cells apoptosis pathway. So, we believe that the current introduction is well introducing what we want to emphasize in manuscript.

Reviewer 2 Report

This study is based on the authors' previous observations. The authors analyzed apoptosis and potential mechanisms using multiple cell lines. Their results suggest a caspase-8 dependent apoptosis may be involved in C5-indued cell death. The study is well designed and the results are sound. The manuscript is also well prepared. There are some minor points may be considered to improve the study.

Fig. 2A. The authors need to show levels of Bcl-XL, BAX, and BID in H1299 and A549 cells. Fig. 3B. How about the Flag-Bcl-2 treatment in Ramos cells? Fig. 4A. Which cell line was used? Ramos or A549? Fig. 4C. How about the effects of Z-LEHD-FMK treatment on A549 cells? Fig. 5B. How about the effects of Flag-FLIP on Ramos cells?

Author Response

We thank you for allowing us to submit to International Journal of Molecular Sciences a revised draft of our manuscript entitled, “C5, a cassaine diterpenoid amine, induces apoptosis via the extrinsic pathways in human lung cancer cells and human lymphoma cells.”. We also appreciate you and each reviewer for the time and effort devoted to providing valuable suggestions on how to improve our manuscript. We have reflected the detailed proposals you have generously provided. We also hope that the following edits and answers we provide will be satisfactorily addressed to all the issues and concerns you and the reviewer have pointed out.

To facilitate the review of our revisions, the following is a point-by-point response to the questions and comments.

This study is based on the authors' previous observations. The authors analyzed apoptosis and potential mechanisms using multiple cell lines. Their results suggest a caspase-8 dependent apoptosis may be involved in C5-indued cell death. The study is well designed and the results are sound. The manuscript is also well prepared. There are some minor points may be considered to improve the study.

We appreciate you time and effort to evaluate our manuscript. We have been trying to identify the anti-cancer mechanism of C5 in multiple cancer cells. We confirmed that C5 works better in certain cancer cell lines and conducted research to find out clear mechanism. We have revealed that C5 affects cell death in cancer cells through extrinsic cell apoptosis pathway and thank you for your positively assessment of these finding results. We are very grateful for your various comments and inquiries for our manuscript. We would like to respond to your comments and questions. And we hope our answers will satisfy you.

Fig. 3A. The authors need to show levels of Bcl-XL, BAX, and BID in H1299 and A549 cells.

We agree with you that we should show the expression of other apoptotic proteins. We have added the result of western blotting to figure 3A. However, we showed that the Bcl-2 expression level is important for the C5-induced apoptosis in figure 3B. Therefore, we believe that the added results little significant to the overall results of our manuscript.

Fig. 3B. How about the Flag-Bcl-2 treatment in Ramos cells?

This structure was made in the same way as Flag-FLIP. We prepared Bcl-2 DNA amplified by polymerase chain reaction and cloned into pCMV-Tag 2B vector. And transfections were performed using Lipofectamin 3000 reagent. We have this explanation in 'Material and method' chapter, and added an explanation. We are grateful for your comment.

Fig. 4A. Which cell line was used? Ramos or A549?

We used Ramos cell line in figure 4A. We misrepresented in figure legend and thank you for your comment of it.

Fig. 4C. How about the effects of Z-LEHD-FMK treatment on A549 cells?

You have raised an important question. However, we checked that the increase of caspase-8 cleavage by C5 treatment in A549 cells. And given that other data show the same result as the Ramos cell, we can conclude that C5 will control cell apoptosis through caspase-8 in A549 cells.

Fig. 5B. How about the effects of Flag-FLIP on Ramos cells?

We acknowledge that has certain limitations; however, we checked that the binding of caspase-8 and FADD was changed in Ramos cells by C5 treatment. This result fully explain that C5 affects the formation of DISC in Ramos cells and we think the C5 effect will be reduced by FLIP alike as in A549.

Once again, thank you for giving us the opportunity to reinforce our manuscript with your opinions and questions. We have done our best to answer your feedback and hope that these revisions are sufficient for you to accept our submission.